# Deep-Learning-Based Polar-Body Detection for Automatic Cell Manipulation

**DOI:** 10.3390/mi10020120

**Published:** 2019-02-13

**Authors:** Yuqing Wang, Yaowei Liu, Mingzhu Sun, Xin Zhao

**Affiliations:** 1Institute of Robotics and Automatic Information System (IRAIS), Nankai University, No. 94 Weijin Road, Nankai District, Tianjin 300000, China; yuqingwang@mail.nankai.edu.cn (Y.W.); nkliuyaowei@foxmail.com (Y.L.); zhaoxin@nankai.edu.cn (X.Z.); 2Tianjin Key Laboratory of Intelligent Robotics (TJKLIR), Nankai University, No. 94 Weijin Road, Nankai District, Tianjin 300000, China

**Keywords:** cell manipulation, automatic micromanipulation, polar-body detection, deep neural network, somatic cell nuclear transfer

## Abstract

Polar-body detection is an essential and crucial procedure in various automatic cell manipulations. The polar body can only be observed when it is located near the focal plane of the microscope, so we need to detect the polar body during cell rotation in cell manipulations. However, three-dimensional cell rotation by micropipette causes polar-body defocus and cell/polar-body deformation, which have not been discussed in existing image-level polar-body-detection approaches. Moreover, varying sizes of the polar bodies increase the difficulty of polar-body detection. In this paper, we propose a deep-learning-based framework to realize polar-body detection in cell rotation. The detection problem is interpreted as image segmentation, which separates the polar body from the background. Then, we improve U-net, which is a typical convolutional neural network (CNN) for medical-image segmentation, so that the network can be applied to polar-body detection, especially for the detection of defocused polar bodies and polar bodies of different sizes. For CNN training, we also designed a particular image-transformation method to simulate more cell-rotation situations, including cell- and polar-body deformation, so that the deformed polar body in cell rotation would be detected by the proposed method. Experiment results show that our method achieves high detection accuracy of 98.7% on a test dataset of 1000 images, and performs well in cell-rotation processes. This method can be applied to various automatic cell manipulations in the future.

## 1. Introduction

The polar body is an important structure during oogenesis, containing a copy of the genetic information of the oocyte. This structure is located between the cytoplasm and the zona pellucida of the oocyte, and is close to the nucleus [1]. Usually, people use polar-body position to localize the nucleus, since the nucleus is not visible under a bright-field microscope. However, due to cytoplasm occlusion, the polar body can only be observed when it is located near the focal plane. Therefore, the polar body, as well as the oocyte, need to be rotated to the desired position in cell micromanipulations, such as somatic cell nuclear transfer (SCNT), intracytoplasmic sperm injection (ICSC) [2], and polar-body biopsy [1].

A three-dimensional illustration of oocyte rotation is shown in Figure 1. The oocyte is held by a holding micropipette, and rotated by the injection micropipette. Usually, the rotation process can be divided into two stages: In the first stage, the polar body is rotated from being invisible to visible. At the end of this stage, the polar body is located near the focal plane. In the second stage, the polar body is rotated to the desired location, such as 2 or 4 o’ clock. During the whole rotation process, there are three difficulties in polar-body detection, as shown in Figure 2: First, the polar body is invisible or defocused most of the time at the first stage. The visual detection of the presence of a polar body or detection in a defocused image is much more difficult than common detection. Second, due to contact with the injection pipette during cell rotation, large-cell deformation is usually generated, easily causing the apparent deformation of the polar body and leading to false detection. Third, polar bodies in different developmental states have different sizes, which requires the detector performing well in a different scale.

In previous methods, polar-body detection was studied as a detection problem at the image level. Leung et al. [3] utilized image binarization and a circle-fitting algorithm to determine the presence of a polar body. Wang et al. [4] introduced a texture-based method to obtain the position of polar bodies of different animals. Wang et al. [5] employed image morphology and ellipse fitting to detect the polar body of a mouse oocyte. Chen et al. [6] introduced a machine-learning method into polar-body detection that applied boundary-curvature information to predict the possible positions of the polar body, and then applied a support vector machine (SVM) algorithm [7] for classifying the image patches in the possible positions.

There are some limitations to these approaches to detect a polar body during cell rotation. Most of the image-level polar-body detection methods focused on image samples with clear polar bodies, but cannot deal with other situations, such as defocusing or deformation. These methods usually employed shape information to predict the position, and used texture information to verify the prediction, due to the assumption that the polar body was a static object with a fixed shape and texture. However, shape-based detection methods, such as circle- or ellipse-fitting, are not suitable for the rotation process, since polar-body deformation is generated during cell rotation. Meanwhile, texture-based detection methods are not reliable for cell rotation either, since texture varies when a polar body is out of focus. Therefore, polar-body detection is still an unsolved problem on the experiment level, especially when the polar body is not obvious in the view.

Nowadays, deep learning has had great success in computer-vision tasks. Convolutional neural networks (CNN), which is a typical category in deep neural networks, use stacks of convolution layers to extract image features and transform data into very high dimensional and nonlinear spaces. Different CNN structures are shown to be very powerful in image-classification, -detection, and -segmentation tasks [8,9,10,11,12,13]. Deep-learning models have also become increasingly popular in biomedical-image processing, for example, dealing with images with different modalities such as CT, MRI, X-ray, and RGB [14,15,16]. Among these models, U-net [17] was designed to realize end-to-end biomedical-image segmentation. It can effectively segment touching objects with very few annotated images for training.

In this paper, we propose a deep-learning framework to realize cell rotation-oriented polar-body detection. Instead of using shape or texture as feature detector, our method learns the difference between a polar body and the background. In order to separate the two parts, the detection problem is interpreted as image segmentation. Specific configurations were designed for the three problems that arise from cell rotation. In order to overcome the out-of-focus problem, which is a convolving process, we utilized a CNN as the main feature detector, and chose U-net [17] as the base framework for polar-body segmentation. Then, we improved the U-net by introducing kernels of different sizes for each layer to realize the detection of polar bodies in different sizes. CNN training needs to collect annotated images in different situations. Besides real experimental images, we also designed three kinds of image transformations to enlarge the dataset and simulate more cell-rotation situations, including cell- and polar-body deformation, so that the deformed polar body in cell rotation could be detected. Extensive experiments were performed to validate the effectiveness of the proposed method. The method had 98.7% accuracy on polar-body prediction for 1000 real porcine-oocyte images. The detection results in the whole rotation process with different oocytes show that our method overcomes the defocus and cell-deformation problems very well, meeting the requirement of automatic micromanipulations.

The rest of the paper is organized as follows: First, the polar-body-detection method is proposed in Section 2. Then, experiment results are illustrated in Section 3. Finally, the paper is concluded in Section 4.

## 2. Deep-Learning-Based Polar-Body Detection

In this section, the proposed method is described in detail. The base segmentation network is illustrated in Section 2.1, which was designed to overcome the defocused-image problem, and which generalizes to polar bodies with different sizes and shapes. Data augmentation for network training is described in Section 2.2. Particular image transformation was utilized to simulate more cell-rotation situations and automatically augment the dataset. Finally, the overall flowchart of the polar-body detection method is described in Section 2.3.

### 2.1. Segmentation Based on Convolutional Network

Instead of learning the shape or texture as the feature detector, we propose to learn the difference between polar body and background. The problem is interpreted as image segmentation, which segments the polar body from the cell. In optics, microscopic imaging and defocusing are defined as the convolution of the ideal image and point spread function (PSF), which motivates us to introduce a convolutional neural network into polar-body segmentation. We chose U-net, which is a typical CNN for medical-image segmentation, as the base framework, and improved the network for polar bodies in different sizes. By using the improved U-net, we predicted the possibility that each pixel belongs to the polar body region, instead of the original 0–1 pixel-wise classification. This design is more in line with a human-cognitive system, i.e., it would predict higher possibility when the polar body is obvious, and lower possibility when the object is not.

#### 2.1.1. Network Architecture

The architecture of the designed CNN for segmentation is illustrated in Figure 3. The input of the network is the origin image. The output of the network is the segmented map in which the value of each pixel represents the probability that the pixel belongs to the polar-body region. The network includes many convolutional layers that are structured into nine modules represented by the blue rectangles. The modules are critical for detecting features.

Since the polar body is a local and small structure, which is quite different from the large and salient object of the image, only using traditional convolutional layers is not enough to segment the polar body. In order to realize the detection of polar bodies in diverse sizes, we utilized diverse kernels for each module. Thus, we introduced the inception module in the Inception-v3 [18] network into U-net, as the module contains multiple small kernels that can scale to different sizes. The details of the module are shown in the I1 box in Figure 3. The input data, which are represented in the base rectangle (array with shape represented by height×width×channel), are fed into four branches: three 1×1 convolution layers and one pooling layer. The 1×1 convolution is used to reduce the channel dimension while maintaining the layer’s spatial information (height×width). Then, different convolutional kernels, sized 3×3, 1×1, 1×3, 3×1 are used to extract image features of objects in different scales. Finally, the data processed by the kernels are concatenated together in the channel dimension and serve as the output layer of this module. Modules I2–I9 have the same structure.

The overall network has a contraction–expansion architecture in which the first four modules encodes the content information, while the last four modules decodes the location information. From I1 to I5, the spatial size (height×width) of the network layer reduces and the channel increases, which means more filters are used to detect content information. The red arrows in Figure 3 represent max-pooling, which can reduce the spatial size of the layer and extract the semantic feature. From I5 to I9, regularity is the opposite, and the spatial size is recovered to the original input-image size. Green arrows represent an upconvolution operation that operates on the opposite direction of the convolution. By using this operation, the network restores the image features and enlarges layer size. Segmentation requires precise location information that is preserved in lower-level image features. Therefore, to combine the low- and high-level image features, skip connections (gray arrows) are utilized to concatenate the symmetrical layers of both the convolution layer and the upconvolution layer of the same level.

#### 2.1.2. Loss Function for Network Training

Deep-learning methods require a loss function that represents the optimization direction. In this paper, we designed two kinds of loss functions to train the neural network: segmentation loss and a classification loss.

The first is a segmentation loss called dice coefficient, which is a global loss for the whole image and used to measure the degree of overlapping between ground truth and our predicted result. In Equation (Equation 1), Mask represents the predicted segmentation mask of the network, and Target represents the corresponding reference mask, i.e., the ground truth. The overlapping of the predicted mask and the ideal mask is measured by the ratio between the intersection and union of two masks. The value of this coefficient is between 0 to 1, the larger, the better. However, for network optimization, we need to minimize loss, so the ratio is subtracted by 1.
(1)$$Dice\_loss=1-\frac{Mask⋂Target}{Mask⋃Target}$$

The second is a classification loss, which is a local loss to measure the correctness of each pixel, and maintain the fineness of the segmented map. The predicted value for each pixel can be considered as the probability that the pixel belongs to the polar-body region, i.e., 1 represents the pixel is predicted as foreground, and 0 represents the pixel that is predicted as background. To realize the classification task, we used categorical cross-entropy loss. As shown in Equation (Equation 2), *N* represents the number of pixels in the image, $x_{i}$ represents the value for each input pixel, and $y_{i}$ represent the expected ground-truth value for this pixel. *h* represent the operation of the network.
(2)$$Cate\_loss=\frac{1}{N}\sum_{i=0}^{n}y_{i}log\left(h_{\theta }\left(x_{i}\right)\right)+\left(1-y_{i}\right)log\left(1-h_{\theta }\left(x_{i}\right)\right)$$

Both losses measure the difference between the predict mask and the ideal target mask. Dice_loss focuses on the outline of the mask, while Cate_loss measures the correctness of all the pixels of the image, no matter if it is in the outline or not. The pixel-level measure helps us to realize more accurate segmentation. To balance the two losses, we also designed different weights (1 and beta) for each loss, and combined them to train the whole network. Through an experiment, we chose the coefficients for Dice_loss and Cate_loss to be 1 and 0.5, respectively, as shown in Equation (Equation 3). Based on Total_loss, we can use a back-propagation algorithm to train the network.
(3)Total_loss=Dice_loss+β×Cate_loss

### 2.2. Data Collection and Augmentation for CNN Training

During cell rotation, polar bodies can be observed at different positions in various situations. Especially when the injection micropipette comes into contact with the oocyte, cell deformation generates and finally leads to polar-body deformation. This problem is hard to overcome through network design. Usually, studies improve algorithm performance by data augmentation for network training. In this paper, we designed three particular kinds of image-transformation methods to simulate more situations in the rotation process.

First, two kinds of image-transformation methods were utilized to simulate oocyte rotation without deformation. For oocytes rotating in the focal plane, we used random-image rotation with angles ranging from −90 to 90 to transform the image. For oocytes rotating vertically to the focal plane, image-axis flipping was used to transform the image.

In addition, elastic transform [19] was applied to simulate more deformation situations. The transform was created by generating a displacement matrix of pixels, i.e., computing a new target location with respect to the original location for every pixel. The process of the elastic transform is listed below:
Generate random displacement fields for each pixel:
(4)Δx(x,y)=rand(−1,+1)
(5)Δy(x,y)=rand(−1,+1)Convolve displacement fields with a Gaussian of standard deviation σ and mean value α:
(6)$$\Delta x^{\prime }\left(x,y\right)=\alpha \times Gaussian\left(\sigma \right)*\Delta x\left(x,y\right)$$
(7)$$\Delta y^{\prime }\left(x,y\right)=\alpha \times Gaussian\left(\sigma \right)*\Delta y\left(x,y\right)$$Generate deformed image according to the new displacements on original image:
(8)$$deform\left(x,y\right)=origin(x+\Delta x^{\prime }\left(x,y\right),y+\Delta y^{\prime }\left(x,y\right))$$

Examples of the three transformations of the same image are shown in Figure 4.

### 2.3. Polar-Body Detection Process

The polar-body detection method is divided into two parts: CNN training, and polar-body position prediction. In training stage, the network operates on the collected and augmented dataset, and uses the loss values as feedback to update its parameters, until the loss value reaches a lower value, which means the network has been trained well.

Figure 5 shows the flowchart of the prediction stage. Detailed steps are listed as follows:Feed the oocyte image (referred to as the first picture in the right) into the trained network; then, a segmented map in black and white is acquired (the second picture).Perform nonmaximum suppression to obtain the most area of the polar body possible. Most of the time, the segmented map has multiple connected regions that represent all possible areas of the polar body. Nonmaximum suppression is used to suppress the areas of lower possibilities. In non-maximum suppression, two constraints are utilized. First is the maximum value in the region. We set the threshold to 0.5 and ignored the regions with a max value lower than 0.5, as the pixel value of the segmented result represents the possibility. Second is the threshold of pixel numbers for the area. After that, areas of the satisfied regions are calculated.Judge if there is a satisfied region. If yes, calculate the mean value of all pixel locations in this region as the center of the polar body; otherwise, put out that there is no polar body in the image.

## 3. Experimental Results

### 3.1. Dataset and Platform

In the experiment, an inverted microscope (IX-51, Olympus, Tokyo, Japan) with a CCD camera (W-V-460, Panasonic, Osaka, Japan) was utilized to capture the image data for CNN network training. Data collection follows two guidelines: On the one hand, we collected images from various oocytes that contained polar bodies of various shapes and sizes. We used 40 oocytes to build the dataset, and samples are shown in Figure 6a. On the other hand, we sampled a series of oocyte images in the first and second stages of oocyte rotation, respectively. For each oocyte, we collected 10 images in the first stage and 40 images in the second stage on average. The first row of Figure 6b shows examples of oocyte images in the first rotation stage, in which some images of the polar bodies were blurred due to defocusing. The first row of Figure 6c shows examples of oocyte images in the second rotation stage, in which the polar bodies point to different orientations, some of them are deformed due to the contact with the injection pipette. We also collected 1500 images without the polar body as negative samples, as shown in the first row of Figure 6d. The training dataset was collected under different light intensities.

In total, 3500 oocyte images were captured. Then, the corresponding ideal masks were manually annotated using Photoshop. As shown in the second row of Figure 6a–d, the boundary of the mask was very precise, corresponding to the original image. In the mask image, the background region is marked as black, in which pixel value equals 0; the polar-body region is marked as white, in which pixel value equals 1. Furthermore, three transformations were utilized to augment the dataset. Finally, the dataset was augmented from 3500 pairs of training images to 14,000 pairs.

To implement the CNN network, we used Keras [20] based on the Tensorflow [21] backend as the coding platform, which is a typical open-source software library for deep learning. The algorithm was running on an Intel I5 processor with 8 GB RAM.

### 3.2. Polar-Body Detection Results

In order to test the performance of our method, we tested 1000 oocyte images from oocyte rotation by using 74 porcine oocytes. The oocytes were collected in different developmental periods. In 1000 images, there are 800 oocyte images with visible polar bodies, and 200 images without a polar body. Given an oocyte image, the algorithm judges if there is a polar body, and then returns the polar-body position if the answer is yes. The quantitative metric is defined as follows: If the polar body is not visible in the image, the prediction of “no position” is regarded as correct; if the polar body is visible in the image, we calculated the mean value of the predicted area and the ground truth. Mean value is calculated as the mean of all pixel locations in this region. If the pixel difference is less than 10 pixels, the prediction could regarded as correct. Other results are regarded as false predictions. The algorithm can successfully identify the presence of a polar body 987 times in these 1000 testings, achieving 98.7% accuracy (ACC). Average detection time was 0.1 s. Table 1 shows the statistical results of polar-body detection. In 13 false detection samples, six of them had wrong predicted positions, and seven of them were predicted as nonexisting.

In 800 images with polar bodies, there were 128 images with defocused polar bodies, 103 images with deformed polar bodies. We also select 74 images from different oocytes. The polar bodies of the oocytes have different sizes and various shapes. Figure 7 shows typical detection results of these three kinds of difficulties. For each detection result, top row shows the original oocyte images with the polar body marked in red rectangles; the bottom row shows the corresponding segmented map in which the white region represents the predicted region of the polar body.

As shown in Figure 7a, when the polar body stays out of the focal plane, it looks smaller than usual, such as the first oocyte image, or its boundary is blurred, such as the second and third oocyte images, even it is difficult for human to distinguish between polar body and background only by texture. However, our method performed well in a defocus situation, with correct and clear segmented maps. In 128 image samples, 126 polar bodies were successfully detected, achieving a success rate of 98.4%.

Similarly, we tested the performance of our method in a deformation situation. As shown in Figure 7b, polar bodies are deformed and squeezed into the cytoplasm due to contact with the injection pipette. The segmented maps show that our method also performs well. Accuracy is 97.1% in 103 image samples.

In order to test the versatility of the proposed method, we randomly chose one image from each of the 74 oocytes. As shown in Figure 7c, there were differences in the positions, sizes, and shapes of the polar bodies, and in image brightness as well. The algorithm successfully detected 73 polar bodies in 74 images. Accuracy was 98.6%. All detection results are summarized in Table 2.

### 3.3. Method Comparison

In Table 3, we compare the proposed method with previous methods in References [3,4,5,6]. We compare the methods in the following aspects:

Accuracy and Testset: Accuracy of the methods on the testset and the size of testset.

Oocyte Type: Types of the oocytes applicable to the method. The polar body in porcine oocyte is smaller than the mouse, which means it is more difficult to deal with the porcine oocyte.

Defocus Availability: whether the method can overcome the defocus problem during cell rotation.

Deformation Availability: If the method can overcome the deformation problem during cell rotation.

Table 3 illustrates the detailed comparison results. The accuracy of our method is the highest in all methods, and this method can be applied to the porcine oocyte, whose polar body is more difficult to detect. Meanwhile, different from previous methods, our method performed well in both defocus and deformation situations, which makes it more suitable for real-experiment applications.

A comparison with traditional standard image-segmentation methods is shown in Figure 8, Each row represents a sample of the typical states of the polar bodies: the polar body in a clearly visible state (Row 1), the polar body in a defocused state (Row 2), the polar body in a deformation state (Row 3), and the polar body in different sizes compared to the first row (Row 4). In the four situations, our method can segment polar bodies very well. For the polar body in a clearly visible state, standard algorithms can segment the polar-body part, but the cell part and micropipette part are also segmented as the foreground. For the polar body in other situations, the standard segmentation algorithm cannot recognize the polar body and even segment it as the background most time. In summary, these methods can only segment the obvious and saliency part as the foreground, but cannot separate the polar body from this part.

The training time of our model on the collected dataset was six hours. However, for application, we only consider prediction time when the model has been trained well; our method cost 0.10 s for each image on the GPU server. The time comparison for all methods mentioned in this paper is illustrated in Table 4. We measured the average time cost for one image in seconds. Previous polar body-detection methods time cost are in the range of 0.13–0.20 s, while segmentation time costs using traditional image-processing methods are in the range of 0.36–320.97 s. Our method is the fastest between all methods. Usually, for automatic cell manipulation, the observation speed is 10 frames per second, but an industrial computer is slower than our GPU server; therefore, we still need to improve the speed of the algorithm to apply the method in realtime cell manipulations in the future.

### 3.4. Application of Polar-Body Detection in Cell Rotation

The proposed method applies to cell rotation; polar-body-detection results are illustrated in Figure 9. Cell rotation is divided into two stages, then we further analyze the deformation situation in detail in the second stage. In each subfigure, the first row shows the original oocyte image. The second row shows the corresponding segmented map, in which, the brightness of the white pixel represents the confidence of the prediction. In order to make the rotation process intuitive, we also plot the polar body in a 3D view in the third row, with the blue circle representing the position of the polar body.

Usually, the polar body rarely appears directly in the image, when oocyte is aspirated and held by the holding pipette. In the first stage of cell rotation, the images of the polar body change from invisible to defocused, then to focused. As shown in Figure 9a, there is no polar body in the first image, since it is out of the focal plane. Correspondingly, our method has predicted the whole image as background without the polar body. In the second image, the polar body is rotated closely to the focal zone and partially visible. Our method has represented the weak possibility of the polar body by the gray region in the segmented map. Subsequently, the polar body gets closer to the focal plane and becomes clearer from left image to right. The corresponding prediction result in the segmented map becomes brighter and more obvious.

In the second rotation stage, the polar body is rotated to the desired position in the focal plane. As shown in Figure 9b, our method segments the images with precise locations, when the polar bodies lie in different positions along the time series. The polar body is often deformed due to contact with the injection micropipette at this stage. Figure 9c shows the images with the deformed polar bodies frame by frame. When the shape of the polar body varies, the corresponding segmented map also varies, but still shows the correct position of the polar body.

Figure 10 shows detection results of other oocytes in the rotation process. Experiment results show that our method is invariant to the position, size, and shape of the polar body and image brightness. More application results on different cells are illustrated in the Appendix A.

## 4. Conclusions

In this paper, we proposed a framework to realize polar-body detection for automatic cell manipulation. Different from previous image-level detection methods, we aim to overcome problems in a real cell-rotation process, including out of focus, deformation, and different sizes of polar bodies. In our method, we first converted detection into image segmentation with a deep convolutional network, and then improved the network by introducing small and different-sized convolutional kernels, so that the method could be applied to detect defocused polar bodies and polar bodies in different sizes. Moreover, we designed three kinds of image-transformation methods to augment the training dataset for the network and simulate more cell-rotation situations, so that the deformed polar body in cell rotation would be detected.

Extensive experiments were performed on 1000 images of 74 porcine oocytes. The accuracies on defocused images, deformation images, and images of oocytes in different development states are 98.4%, 97.1%, and 98.6%, respectively. The overall accuracy of 1000 images was 98.7%, higher than all pervious methods. Our method also performed well during cell rotation, which meets the requirement of automatic biological cell-manipulation tasks. Our method costs 0.10 s for each image on the GPU server, while the industrial computer that is now used for cell manipulations is slower than the GPU server. Therefore, we still need to improve the speed of the algorithm and add GPU into the industrial computer to apply the method in real-time cell manipulations in the future. 

## Figures and Tables

**Figure 1 micromachines-10-00120-f001:**
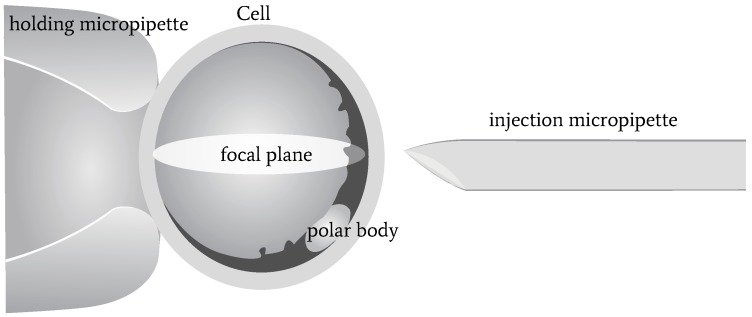
Three-dimensional illustration of oocyte rotation.

**Figure 2 micromachines-10-00120-f002:**
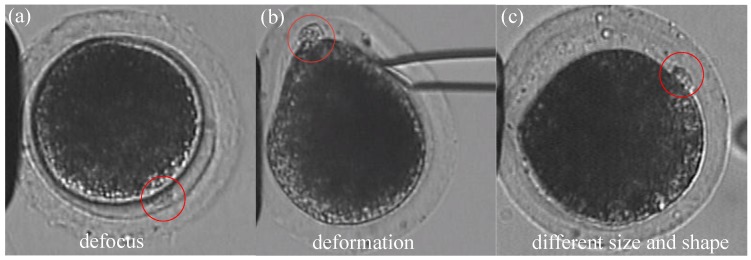
Three typical problems for polar-body detection in oocyte-rotation processes. Polar bodies are indicated by red circles. (**a**) Polar body in defocused situation. (**b**) Polar-body deformation caused by micropipette. (**c**) Polar body in different sizes (height and width of the image are 256 pixels in the whole paper, with 16 pixels representing 10 μm. Scale bars were added in (**a**).

**Figure 3 micromachines-10-00120-f003:**
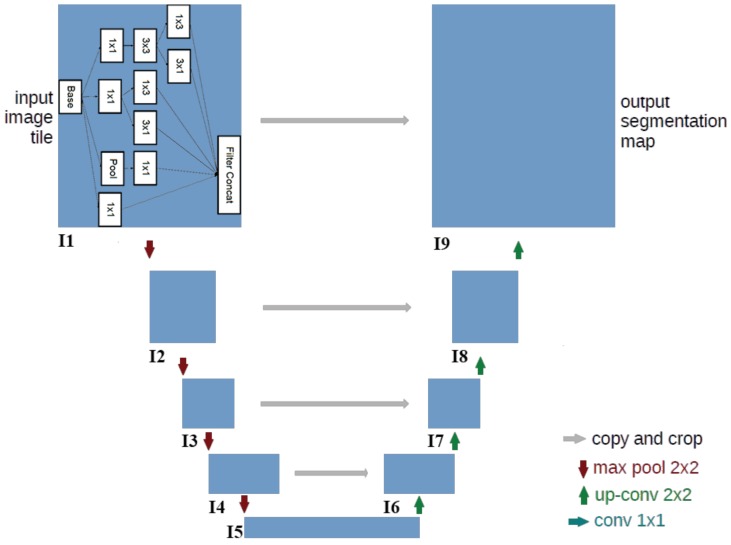
Segmentation-network architecture.

**Figure 4 micromachines-10-00120-f004:**
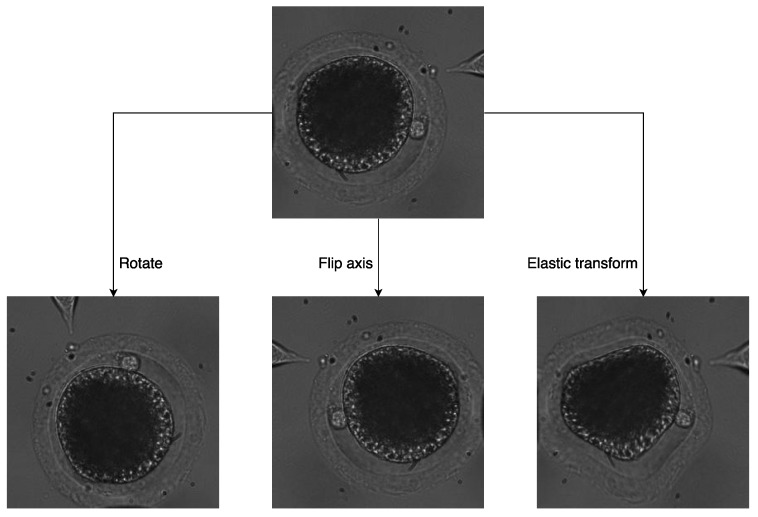
Three data-augmentation methods.

**Figure 5 micromachines-10-00120-f005:**
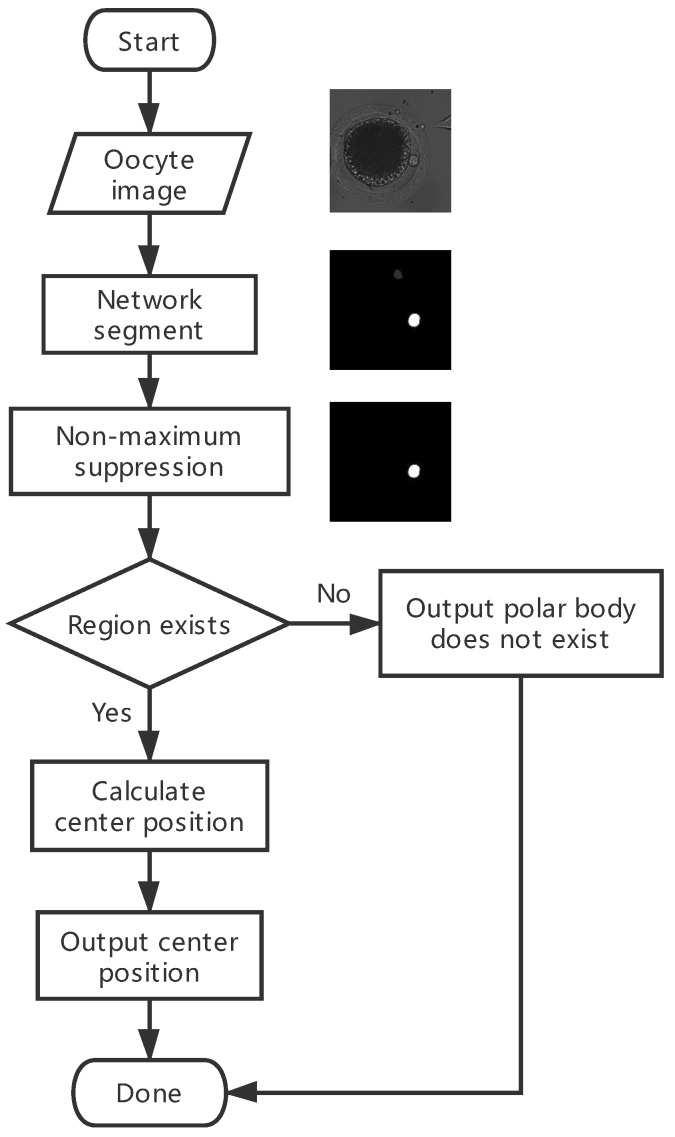
Flowchart of polar-body detection process.

**Figure 6 micromachines-10-00120-f006:**
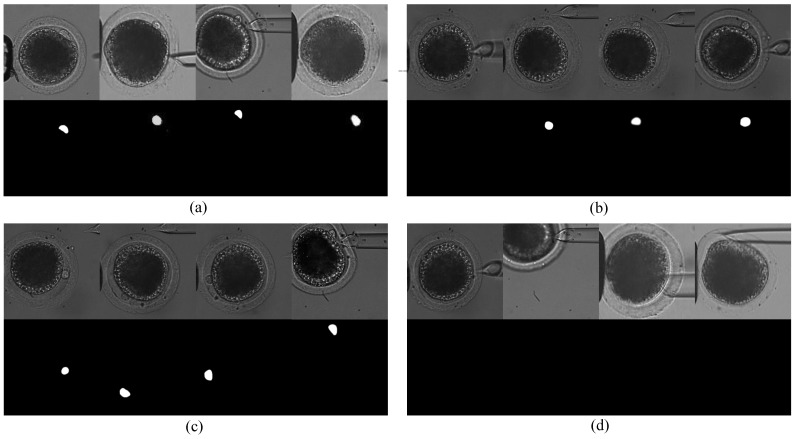
Examples of training data. (**a**) Polar-body images of different oocytes. (**b**) Polar-body images of the same oocyte in the first rotation stage. (**c**) Polar-body images of the same oocyte in the second rotation stage. (**d**) Oocyte images without a polar body as negative samples. Each union contains two rows. First row, oocyte images; second row, corresponding masks. White regions show the regions of the polar bodies.

**Figure 7 micromachines-10-00120-f007:**
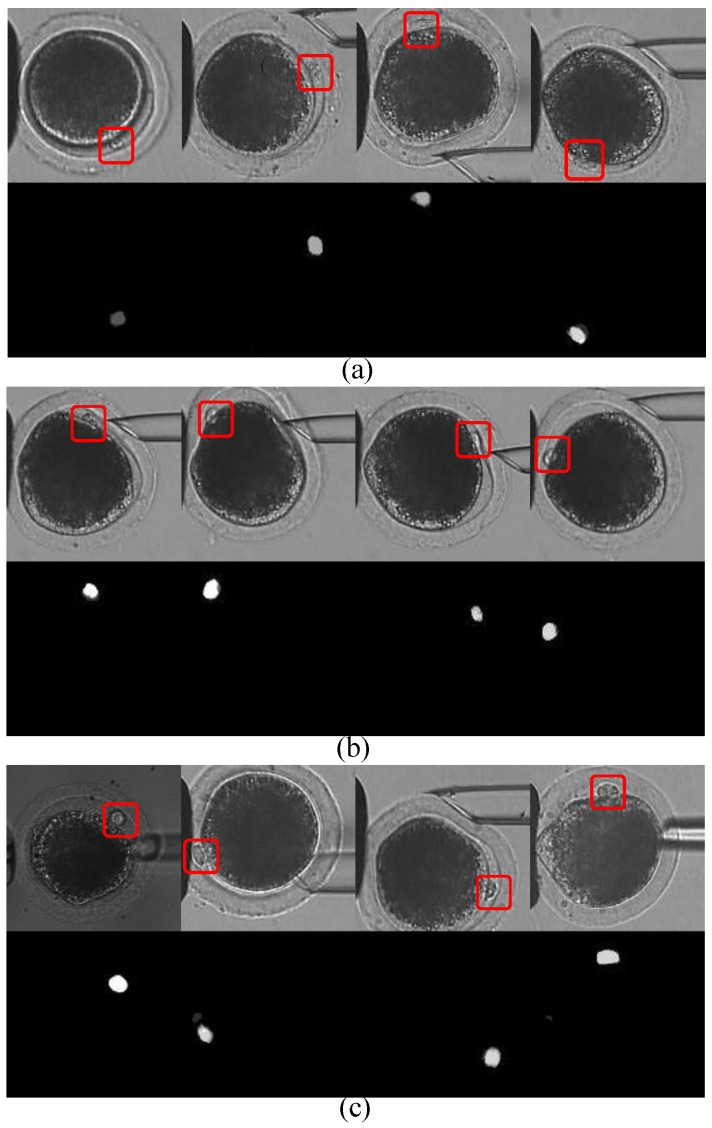
Polar-body detection results in (**a**) defocus situation; (**b**) deformation situation; and (**c**) different oocytes. For each detection result, top row shows the original oocyte image, bottom row shows corresponding segmented map. Polar bodies are marked by red rectangles for better view.

**Figure 8 micromachines-10-00120-f008:**
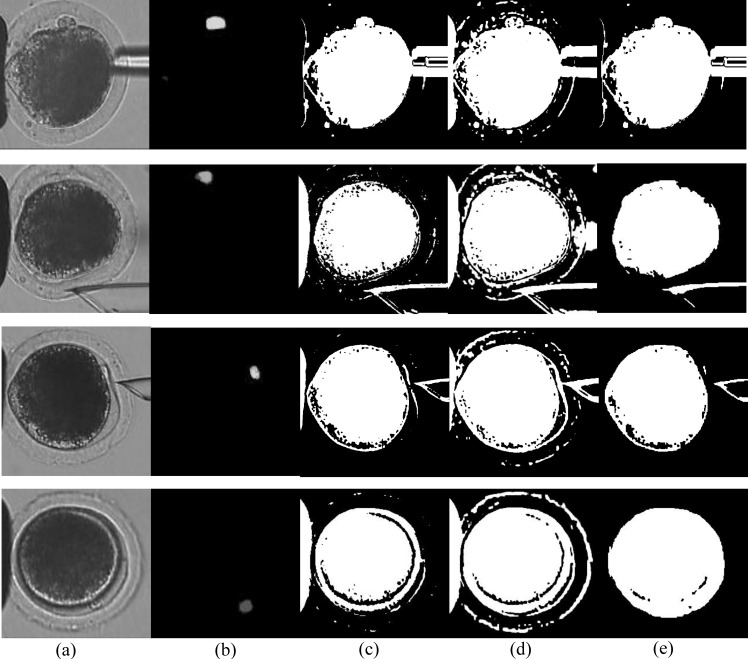
Comparison with traditional standard image-segmentation methods. Each column represents segmentation results of four image-processing methods: (**a**) original image, (**b**) our method, (**c**) Otsu thresholding algorithm [22], (**d**) Gibbs algorithm [23], (**e**) GrabCut algorithm [24].

**Figure 9 micromachines-10-00120-f009:**
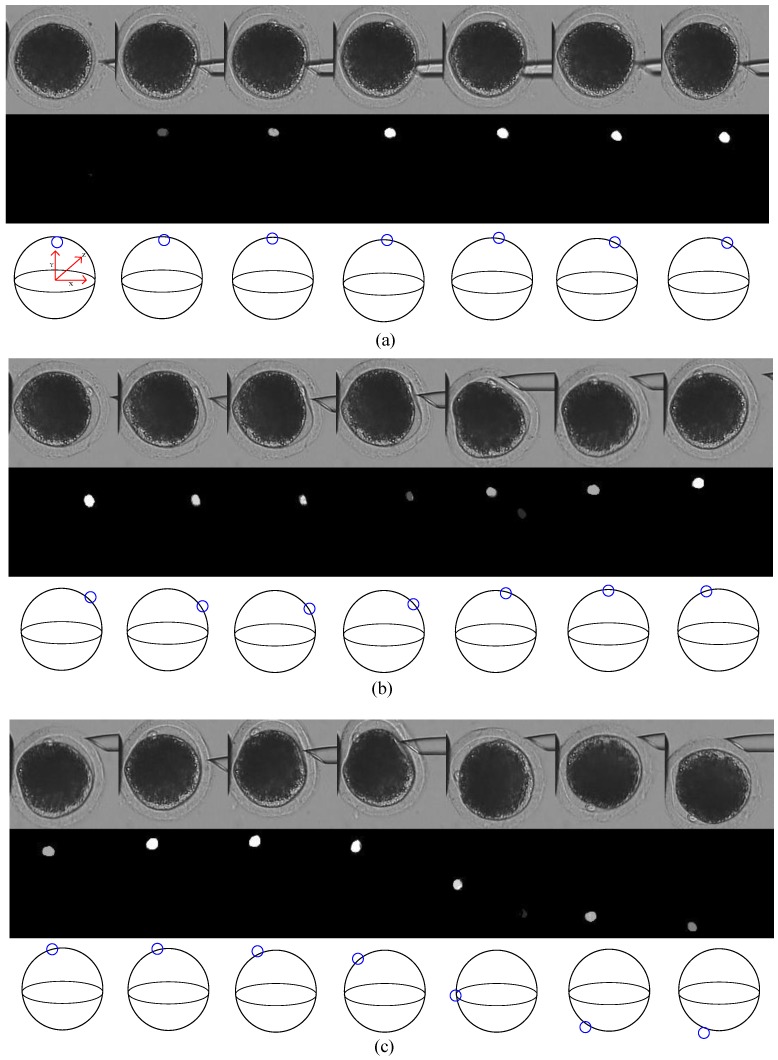
Polar-body-detection results during cell rotation. (**a**) In the first rotation stage, the polar body is from invisible to visible, from blurred to clear. (**b**) In the second rotation stage, the polar body is rotated to the desired position in the focal plane. (**c**) In the second rotation stage, the polar body is often deformed due to contact with the injection micropipette. For each process, the first row represents the original oocyte image. Polar bodies are in different positions along the time series of the rotation process. The second row represents the corresponding segmented map in which the brightness of the white pixel represents the confidence of the prediction. The third row represents the polar body in a 3D view in which the position of the polar body is marked by a blue circle.

**Figure 10 micromachines-10-00120-f010:**
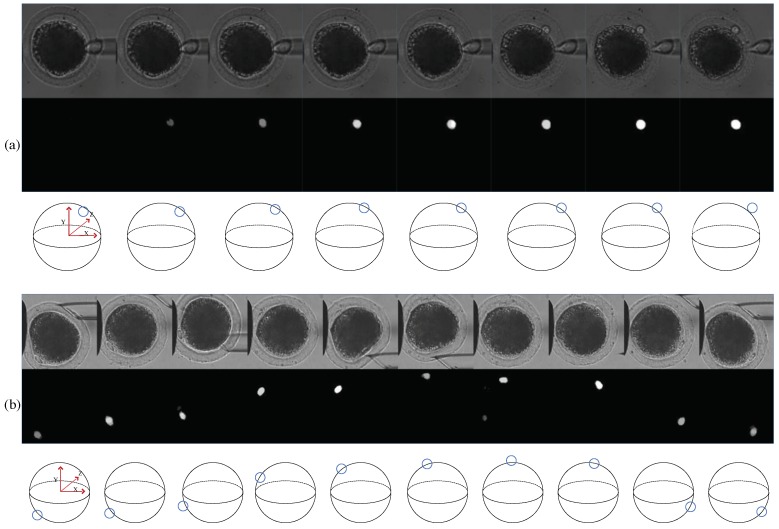
Detection result of two different oocytes during the rotation process. (**a**) Polar body from invisible to visible. (**b**) Polar body rotated around the cell circle.

**Table 1 micromachines-10-00120-t001:** Test accuracy (ACC) of polar-body detection.

Class	Test Samples	Correct Predictions	ACC
Positive	800	787	98.3%
Negative	200	200	100%
Summary	1000	987	98.7%

**Table 2 micromachines-10-00120-t002:** Test accuracy of three typical situations.

Situations	Test Samples	Correct Predictions	ACC
Defocused	128	126	98.4%
Deformation	103	100	97.1%
Different oocytes	74	73	98.6%

**Table 3 micromachines-10-00120-t003:** Comparison of previous polar-body methods and our method.

Methods	ACC	Testset	Oocyte Type	Defocused	Deformation
Leung’s [3]	93%	30	Mouse	No	No
Wang’s [4]	80%	-	-	No	No
Chen’s [6]	96%	8622	Porcine and Mouse	Yes	No
Wang’s [5]	96%	80	Mouse	No	Yes
Ours	98%	1000	Porcine	Yes	Yes

**Table 4 micromachines-10-00120-t004:** Average time cost of methods mentioned in this paper for processing an image.

Methods	Leung’s [3]	Wang’s [4]	Chen’s [6]	Wang’s [5]	Otsu [22]	Gibbs [23]	GrabCut [24]	Ours
seconds	-	-	0.20	0.13	0.36	2.92	320.97	0.10

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
