# Peer review of "Deep-Learning-Based Polar-Body Detection for Automatic Cell Manipulation"

_micromachines, 2019, doi:10.3390/mi10020120_

Round 1

Reviewer 1 Report

This paper presents a very interesting and important study for detection of polar body by using a CNN model. The detection was firstly converted as image segmentation. Methods were developed to tackle the difficulties of defocusing, varying sizes, and deformation of the polar bodies during detection. Three kinds of image transformation were designed to augment the training dataset and simulate cell rotation. Experiments were well designed and performed to validate the accuracy and efficacy of the proposed methods for polar body detection. This paper can be accepted if the following questions are well addressed:

(1) In the Introduction, the authors reviewed the literature in polar body detection using traditional image processing; the authors are encouraged to review the techniques of deep learning for image analysis, and to explain why U-Net model is adopted in this study.

(2) Is the training dataset collected under the same light intensity? Will the light intensity influence the detection of the polar body?

(3) What is the image size (height and width) in polar body detection? Please add a scale bar to the figures if possible.

(4) In Section 3.2, the author explained “In 13 false detection samples, 6 of them have wrong predicted positions”, what is the quantitative metric to identify if a predicted position is correct. 

Author Response

We are truly grateful to your and other reviewers’ critical comments and thoughtful suggestions to help us improve our paper. Based on these comments and suggestions, we have made careful modifications on the original manuscript. We hope the new manuscript will meet your magazines standard. Our point-by-point responses to the reviewers’ comments/ questions are uploaded in the Word file.

Reviewer 2 Report

This manuscript reported the cell image process via deep learning algorithm for identifying the polar body during cell manipulation.  My specific comments are as follows.

1.The image process via the deep learning algorithm, which is a standard algorithm in deep learning, shows promising results for polar body identification.  However, some standard image process might work better for the purpose of automatic cell manipulation.    

2.For the practical application of automatic cell manipulation, the calculation speed of deep learning for real-time manipulation might be a weakness for this paper.  I strongly suggest the authors clearly address the present time period for the deep learning process as addressed in this manuscript (at least in the conclusion section).  Also clearly make comparison with the max-allowed time period of image process for the application of the automatic cell manipulation described in this manuscript.  

3.Except above concerns, this is a good research report.   

Author Response

(The authors gave the same response as above.)
